# MALDI Mass Spectrometry Imaging—Prognostic Pathways and Metabolites for Renal Cell Carcinomas

**DOI:** 10.3390/cancers14071763

**Published:** 2022-03-30

**Authors:** Franziska Erlmeier, Na Sun, Jian Shen, Annette Feuchtinger, Achim Buck, Verena M. Prade, Thomas Kunzke, Peter Schraml, Holger Moch, Michael Autenrieth, Wilko Weichert, Arndt Hartmann, Axel Walch

**Affiliations:** 1Institute of Pathology, University Hospital Erlangen-Nuremberg, 91054 Erlangen, Germany; arndt.hartmann@uk-erlangen.de; 2Research Unit Analytical Pathology, Helmholtz Zentrum München–German Research Center for Environmental Health, 85764 Neuherberg, Germany; jian.shen@helmholtz-muenchen.de (J.S.); annette.feuchtinger@helmholtz-muenchen.de (A.F.); achim.buck@helmholtz-muenchen.de (A.B.); verena.prade@helmholtz-muenchen.de (V.M.P.); thomas.kunzke@helmholtz-muenchen.de (T.K.); axel.walch@helmholtz-muenchen.de (A.W.); 3Department of Pathology and Molecular Pathology, University Hospital Zurich, 8091 Zurich, Switzerland; peter.schraml@usz.ch (P.S.); holger.moch@usz.ch (H.M.); 4Department of Urology, Rechts der Isar Medical Center, Technical University of Munich, 81675 Munich, Germany; michael.autenrieth@tum.de; 5Institute of Pathology, Technical University Munich, 81675 Munich, Germany; wilko.weichert@tum.de

**Keywords:** clear-cell renal cell carcinoma, papillary renal cell carcinoma, chromophobe renal cell carcinoma, mass spectrometry imaging, metabolomics

## Abstract

**Simple Summary:**

Renal cell carcinoma (RCC) is the seventh most common cancer type and accounts for more than 80% of all renal tumors. Nevertheless, prognostic biomarkers for RCC are still missing. Therefore, we analyzed a large, multicenter cohort including the three most common RCC subtypes (clear cell RCC (ccRCC), papillary RCC (pRCC) and chromophobe RCC (chRCC)) by high mass resolution matrix-assisted laser desorption/ionization (MALDI) mass spectrometry imaging (MSI) for prognostic biomarker detection. This is a suitable method for biomarker detection for several tumor entities. We detected several pathways and metabolites with prognostic power for RCC in general and also for different RCC subtypes.

**Abstract:**

High mass resolution matrix-assisted laser desorption/ionization (MALDI) mass spectrometry imaging (MSI) is a suitable method for biomarker detection for several tumor entities. Renal cell carcinoma (RCC) is the seventh most common cancer type and accounts for more than 80% of all renal tumors. Prognostic biomarkers for RCC are still missing. Therefore, we analyzed a large, multicenter cohort including the three most common RCC subtypes (clear cell RCC (ccRCC), papillary RCC (pRCC) and chromophobe RCC (chRCC)) by MALDI for prognostic biomarker detection. MALDI-Fourier-transform ion cyclotron resonance (FT-ICR)-MSI analysis was performed for renal carcinoma tissue sections from 782 patients. SPACiAL pipeline was integrated for automated co-registration of histological and molecular features. Kaplan–Meier analyses with overall survival as endpoint were executed to determine the metabolic features associated with clinical outcome. We detected several pathways and metabolites with prognostic power for RCC in general and also for different RCC subtypes.

## 1. Introduction

The incidence of renal tumors has increased in the past few decades worldwide, especially in Europe, the United States and Australia. Renal cell carcinoma (RCC) accounts for more than 80% of renal tumors. RCC is now the seventh most common cancer type [1]. In 2016, the World Health Organization (WHO) defined the three most common tumor types: the clear-cell RCC (ccRCC), the papillary RCC (pRCC), and the chromophobe RCCs (chRCC) constitute 80–90%, 10–15%, and 4–5% of cases, respectively [2].

To predict the course of RCC for patients, a variety of grading systems have been proposed. The four-tiered WHO/International Society of Urological Pathology (ISUP) has been validated for ccRCC and pRCC. The system defines tumor grade 1–3 on the basis of nucleolar prominence. Grade 4 contains, inter alia, the presence of pronounced pleomorphism and/or sarcomatoid differentiation [2,3]. The chRCC shows a favorable prognosis. The 5-year survival rate is 78–100% [4]. Hence, some patients already show metastases at the time of diagnosis [5]. Until now, several grading systems have been proposed, but none of them became widely accepted [6,7,8].

Many additional prognostic markers have been suggested for different RCC subtypes; for example, Programmed Death-Ligand 1 (PD-L1), the mesenchymal–epithelial transition factor (cMET) or Claudin 7 (CLD7) [9,10,11,12,13]. Nevertheless, none of these biomarkers are used in daily routine because of lacking validity [2].

Several studies showed the power of high mass resolution matrix-assisted laser desorption/ionization (MALDI) mass spectrometry imaging (MSI) for tumor discrimination [14,15,16,17]. Moreover, this method identified molecular features, which are associated with clinico-pathological parameters in ccRCC [18,19]. Hence, the aim of this study is to define novel prognostic pathways and metabolites for the three most common RCC subtypes. Therefore, we analyzed 782 patient tissue samples by MALDI-MSI, including 552 ccRCCs, 122 pRCC, and 108 chRCC. We compared the metabolic profile with overall survival (OS) and detected several RCC-specific and also subtype-specific patterns. To the best of our knowledge, this is the first study which analyzed a large multicenter RCC cohort by MALDI-MSI according to OS.

## 2. Materials and Methods

### 2.1. Patient Tissues

Formalin-fixed paraffin-embedded (FFPE) renal tumor samples comprising 552 clear cell renal cell carcinoma (ccRCC), 122 papillary renal cell carcinoma (pRCC) and 108 chromophobe renal cell carcinoma (chRCC) were collected from the archives of the Department of Pathology and Molecular Pathology of the University Hospital Zurich (1993–2013) and of the Technical University of Munich (1996–2014). Tissue microarrays (TMAs) were constructed as described [20]. Tissue cylinders with 0.6 mm (Zurich) and 1.0 mm (Munich) diameter were punched from morphologically representative regions of paraffin donor blocks.

### 2.2. MALDI Mass Spectrometry Imaging

Tissue preparation steps for the high mass resolution matrix-assisted laser desorption/ionization Fourier-transform ion cyclotron resonance mass spectrometry imaging (MALDI-FT-ICR-MSI) analysis was performed as previously described [21]. In brief, FFPE TMAs were sectioned with 4 µm (Microm, HM340E, Thermo Fisher Scientific, Waltham, MA, USA) and mounted onto indium-tin-oxide (ITO)-coated glass slides (Bruker Daltonik, Bremen, Germany) pretreated with 1:1 poly-L-lysine (Sigma-Aldrich, Munich, Germany) and 0.1% Nonidet P-40 (Sigma-Aldrich, Munich, Germany). FFPE sections were adhered by incubating the slide for 1 h at 70 °C, deparaffinized in xylene (2 × 8 min), and air-dried. The matrix solution consisted of 10 mg/mL 9-aminoacridine hydrochloride monohydrate matrix (Sigma-Aldrich, Munich, Germany) in 70% methanol. Spray-coating of the matrix was conducted using the SunCollectTM sprayer (Sunchrom, Friedrichsdorf, Germany). The flow rates were 10 µL/min, 20 µL/min, 30 µL/min for layers 1–3, and layers 4–8 with 40 µL/min, utilizing 2 mm line distance and a spray velocity of 900 mm/min.

Metabolites were detected in negative-ion mode on a 7 T Solarix XR FT-ICR mass spectrometer (Bruker Daltonik, Bremen, Germany) equipped with a dual ESI-MALDI source and a SmartBeam-II Nd: YAG (355 nm) laser. Laser attenuation value is 15%. The laser operated with focus of small (84.2%). The laser operated at a frequency of 1000 Hz utilizing 200 laser shots per pixel with a pixel resolution of 60 µm. Data acquisition parameters were specified in ftmsControl software 2.2 and flexImaging (v. 5.0) (Bruker Daltonik, Bremen, Germany). Mass spectra were acquired in negative-ion mode covering *m*/*z* 75–1000. The three RCC subtypes (ccRCC, pRCC, chRCC) were randomly distributed over 11 TMAs. The MSI experiments of the TMA sections were randomized and measured within a batch. For internal mass calibration, the 9-AA matrix ion signal (*m*/*z* 193.077122) was used as lock mass minimizing scan-to-scan (pixel-to-pixel) variations during the MALDI-MSI measurement. External calibration of the instrument was performed with L-Arginine in the ESI mode. MALDI mass spectra were root mean square normalized with SCiLS (v. 2020b Pro), and picked peaks were exported as imzML files for further data processing and subsequent analysis with the SPACiAL pipeline. SPACiAL pipeline was integrated for automated co-registration of histological and metabolic features as previously described [22]. Briefly, the MALDI mass spectra were root mean square normalized with SCiLS (v. 2020b Pro) and exported as imzML files. An in-house python 3 pipeline was performed for pixel-wise and parallelized peak picking. For each coordinate (i.e., spectrum), the peak picking pipeline began by resampling the mass and intensity values between 75 and 1000 Da with a step size of 0.0005 Da. Intensity values were resampled by choosing the maximum intensity per window. Noise levels were estimated for windows of 10 Da, and all peaks falling below their respective noise level were filtered. After noise filtering, only local maxima were kept as preliminary peaks. Preliminary peaks within each spectrum were merged and aligned. Peaks that occur in less than 0.5% of the spectra were filtered. For image co-registration, the imzML file of picked peaks was used to create a master image of the MALDI measurement region. Additional hematoxylin & eosin (H&E) stained images of the same tissue sections were precisely co-registered onto this image, allowing an exact integration and correlation of molecular MALDI data with morphology data. The digitized and co-registered staining images were scaled to match the exact MALDI resolution and then converted into numerical data without loss of spatial resolution. Regions of interest (tumor) were annotated based on H&E stainings. Mean peak intensities of tumor regions were extracted for each patient and used for prognosis analysis.

Peak annotation was performed using HMDB and METASPACE (http://annotate.metaspace2020.eu/ accessed on 15 December 2021) databases, while allowing M-H, M-H_2_O, M + K-2H, M + Na-2H, and M + Cl as negative adducts with a mass tolerance of 4 ppm (Ion mode: negative, Adduct type: [M-H], [M-H-H_2_O], [M + Na-2H], [M + Cl] and [M + K-2H], mass accuracy ≤ 4 ppm). Pathway analysis was performed with Kyoto Encyclopedia of Genes and Genomes (KEGG) database (http://www.genome.jp/kegg/ accessed on 15 December 2021).

### 2.3. Bioinformatics and Statistical Analysis

Overall survival was defined as the time from primary surgery to death or last follow-up and were calculated using the Kaplan–Meier method and include 95% confidence interval (95% CI) estimates. Survival curves were tested with the log-rank χ2 value. In each case, the cutoff point was optimized with respect to the endpoint. In order to determine the prognostic power for each metabolite, the individual patient metabolite abundances were used to split the cohort into good and poor survivor groups by the application of intensity cutoffs, which were optimized to the clinical endpoint. Cutoff-optimized survival analyses were performed as previously described [23,24] using a Kaplan–Meier Fitter and log-rank test. Cutoff-optimized in this context means that the threshold for low and high abundance of a compound was chosen such that the *p*-value in the resulting Kaplan–Meier curve is minimal. Survival analyses were performed within the R statistical environment including “Survival” package (R Foundation for Statistical Computing, Vienna, Austria), and *p*-values < 0.05 were considered statistically significant.

Box plots were created with GraphPad PRISM v. 5.00 (GraphPad Software, Inc., La Jolla, CA, USA). Statistical significance testing was performed using the Kruskal–Wallis test (alpha = 0.05).

## 3. Results

### 3.1. Sample Description and MALDI-MS Imaging Experiments

Spatial metabolomic was performed for renal carcinoma tissue sections from 782 patients comprising three tumor subtypes—clear cell renal cell carcinomas (ccRCC, *n* = 552), papillary renal cell carcinomas (pRCC, *n* = 122) and chromophobe renal cell carcinomas (chRCC, *n* = 108) (Table 1, Appendix A). The workflow integrated high mass resolution matrix-assisted laser desorption/ionization Fourier-transform ion cyclotron resonance mass spectrometry imaging (MALDI-FT-ICR-MSI) analysis, SPACiAL pipeline for automated co-registration of histological and molecular features, bioinformatics analysis for patient prognosis analysis. Within the mass range of *m*/*z* 75 to 1000, in total, 2111 metabolite peaks were resolved. These metabolic features were used for further statistic and prognosis analysis.

### 3.2. Kaplan–Meier Survival Analysis

To determine whether the metabolic features associated with clinical outcome, we performed Kaplan–Meier analyses with overall survival as endpoint. The result revealed that 240 Kaplan–Meier significant metabolites were identified in chRCC, 108 in ccRCC, and 242 in pRCC (Figure 1). The Venn diagrams showed that 17 Kaplan–Meier significant metabolites were commonly identified in all three renal subtypes (Figure 1). Specific nucleotides (Guanosine monophosphate, Cyclic GMP) indicated strong impacts on patient outcomes in all three renal subtypes. The increased abundance of these nucleotides was associated with poor patient prognosis (Figure 2 and Figure 3). Ribose phosphate is an important intermediate metabolite in the pentose phosphate pathway and in the purine metabolism pathway. Based on the Kaplan–Meier survival analysis, ribose phosphate was associated with unfavorable patient outcome (Figure 4). Kruskal–Wallis test indicate that the abundances of guanosine monophosphate, cyclic GMP and ribose phosphate showed no significant difference in three renal subtypes.

We determined subtype-specific prognostic metabolites. Figure 5 represented Kaplan–Meier significant metabolites specific for subtype chRCC including nucleotide and derivatives, oxidative phosphorylation, acrylaminosugars, pentose phosphates, and numbers of lipids and fatty acids. Kaplan–Meier significant metabolites specific for subtype ccRCC consisted of nucleotides (cyclic AMP, cytidine diphosphate and uridine monophosphate), glutathione disulfide and a lysophosphatidic acid (Figure 6). Furthermore, Kaplan–Meier significant metabolites specific for subtype pRCC comprised glucosamine as aminosugar and 2-sulfinoalanine from cysteine and methionine metabolism (Figure 7). Appendix A summarized the mass intensities and annotations of the prognosis metabolites described in this study.

## 4. Discussion

Several previous studies analyzed different RCC subtypes using MALDI. The diagnostic power of MALDI for tumor discrimination of several entities has already been shown. Chinello et al., identified a peptide cluster, which can discriminate 33 patients with ccRCC from 29 healthy controls using the ClinProt/MALDI-TOF technique [25]. Junker et al., analyzed 27 patient tissue samples with RCC and detected stage-related protein alterations using MALDI-TOF-MS/MS [26]. Zhang et al., demonstrated that MALDI enables discrimination of renal Oncocytoma (rO) from RCC subtypes and normal kidney tissue in 81 banked frozen human tissue samples [14]. Kriegsmann et al., analyzed 71 chRCC and 64 rO by MALDI and achieved an accuracy of 89% in tumor discrimination [16]. Recently, Prade et al., showed a synergism effect in tumor subtyping by using MSI and morphometry combined [27].

Furthermore, some MALDI studies concentrated on the prognostic power, especially with respect to ccRCC. For example, Steurer et al., performed MALDI analysis on 789 RCC related to their clinicopathological features. They found significant associations with tumor stage, Fuhrman grade and presence of lymph node metastases in the ccRCC subgroup. For the other RCC subgroups, no significant associations were detected [18]. As mentioned above, tumor grade represents one of the most important parameters to evaluate ccRCC progression. Therefore, Stella et al., performed a study which uncovered protein alterations associated with different ccRCC grade lesions by MALDI-MSI. The data highlighted this method to be able to discriminate among different grades of ccRCC, and thus to have a prognostic power [19].

Up to now, comparable studies which focus on the prognostic power of MALDI-MSI using large multicenter cohorts including the three most common RCC subtypes are still missing. Therefore, our study concentrated on the prognostic relevant pathways and metabolites for all RCCs and on the tumor-specific features.

First of all, it is important to mention that only 17 Kaplan–Meier (KM) significant metabolites were detected in all three tumor subtypes. This indicates that each of the three RCC subtypes shows a quite unique metabolic environment. Metabolic differences associated with kidney origin could occur for chRCC that originates from intercalated cells of the collecting duct, but ccRCC and ppRCC metabolic variability are probably due to other causes, because both subtypes originate from the proximal tubules [28].

In specific, nucleotides (Guanosine monophosphate, Cyclic GMP) indicated strong impacts on patient outcomes in all three renal subtypes. The increase of these nucleotides was associated with poor prognosis.

Particularly, cyclic guanosinmonophosphate (cGMP) is a very interesting marker in this context. Cyclic guanosine monophosphate-adenosin monophosphate (cGAMP) synthase (cGAS) is a cytosolic DNA sensor, which activates innate immune response. Normally, self-DNA is localized in the nucleus and mitochondria in an eukaryotic cell. DNA is a danger-associated pattern, if self-DNA is present in the cytosol [29,30]. cGAS binds to double-stranded DNA irrespective of the DNA sequence [31]. This mechanism plays an essential role in cellular senescence, which is a natural barrier to tumorigenesis [32]. cGAS catalyzes the conversion of guanosintriphosphate (GTP) and adenosintriphosphat (ATP) into 2′3′-cGAMP. This molecule contains two phosphodiester bonds, one between the 2′-OH of GMP and 5′-phosphate of AMP, and the other between the 3′-OH of AMP and 5′-phosphate of GMP [33,34]. cGAMP is a second messenger that binds and activates the adaptor protein stimulator of interferon genes (STING) [35]. STING is localized on the endoplasmic reticulum membrane and activates inter alia two protein kinases: IkappaB (IkB or IKK) and Tank-binding kinase 1 (TBK1). This leads to an activation of the transcription factors nuclear factor kappa B (NF-κB) and interferon regulatory factor 3 (IRF3). As a result, several immune and inflammatory gene products, such as type I interferons (IFNs) and tumor necrosis factor α (TNFα). The cancer therapy, especially for RCC, has been revolutionized based on the use of immune checkpoint inhibitors with targets such as programmed death 1 (PD-1), PD-L1 and cytotoxic T-lymphocyte-associated protein 4 (CTLA-4).

The role of STING in tumor immunity is still unclear. Several studies demonstrated that STING-deficient mice are less responsive to immunotherapies [36,37]. On the other hand, some studies suggest that STING activation may induce a suppressive tumor microenvironment and contribute to tumor growth and metastasis [38,39].

Ren et al., suggested that the cGMP pathway plays an essential role in regulation and survival of RCC [40]. Furthermore, Msaouel et al., detected that the renal medullary carcinoma (RMC), which is a highly lethal malignancy, is characterized by high replication stress and an abundance of focal copy number alterations associated with activation of the stimulator of the cyclic GMP-AMP synthase interferon genes’ innate immune pathway [41]. These findings support our results and lead to the hypothesis that an enrichment of the cGMP pathway is associated with worse prognosis in all types of RCC.

Furthermore, we detected pathways which seem to be tumor type-specific. For example, the Gluthation metabolism for ccRCC. Recent studies showed that this pathway seems to play an essential role in RCC. Hakimi et al., demonstrated that the Gluthation metabolism is increased in late-stage ccRCC and is associated with worse survival outcomes in ccRCC patients [42]. Gluthation is a reactive oxygen species (ROS) scavenger, which could be a hallmark of RCC [43]. The function of this increased pathway is to counteract damaging ROS. This leads to a better viability and growth of the malignant cells [43].

Moreover, we detected an increased cysteine metabolism as prognostic relevant pathway for pRCC. Cysteine is a Gluthation metabolism-related metabolite, which shows a significant power as prognostic marker in pRCC. Glutathione is a tripeptide thiol antioxidant, which contains the following amino acids: glutamic acid, cysteine, and glycine [44]. Alahmad et al., showed that the metabolome demonstrated a tremendous increase of Gluthation in both forms, reduced and oxidized, in pRCC [45]. Therefore, the increased cysteine metabolism could be an indirect marker for an increase of the Gluthation metabolism.

In contrast to ccRCC and pRCC, Priolo et al., demonstrated that chRCC show a striking decrease in intermediates of the glutathione pathway compared with adjacent normal kidney [46]. These results are comparable with our results. We also did not detect any increase of the Gluthation pathway in this tumor entity. Whereas, we detected an increase of Uridinmonophoshat, for example. These findings suggest that the chRCC differs in his tumor biology from the other two entities to a great extent.

This study demonstrates the ability to measure metabolites in FFPE tissues using MALDI-FT-ICR MSI, which can then be assigned to histology and clinical parameters. Although removal/reduced intensity of hydrophobic molecules during the deparaffinization process of FFPE samples were observed, there were still multiple classes of robust metabolites, not only chemically, but also spatially preserved in FFPE tissue specimens [21,47]. The identification and characterization of metabolite structures are limited by the analytical depth of MSI in terms of coverage and identification capabilities, and the identification of new, previously uncharacterized metabolites is difficult. The term “annotation” implies putative. Although alternative annotations exist for some compounds, most are structural variants of the same compound or priorities can be assessed according to mass accuracy, adduct, or plausibility. Of course, these findings have to be confirmed in future studies, which should focus on all types of RCC, including the rarer subtypes. Finally, we can summarize that MALDI-MSI is a promising approach in detecting novel tumor-specific prognostic markers.

## 5. Conclusions

We detected several pathways and metabolites with prognostic power for RCC in general and also for different RCC subtypes. Therefore, MALDI is a high-potential technique for biomarker detection in several tumor entities.

## Figures and Tables

**Figure 1 cancers-14-01763-f001:**
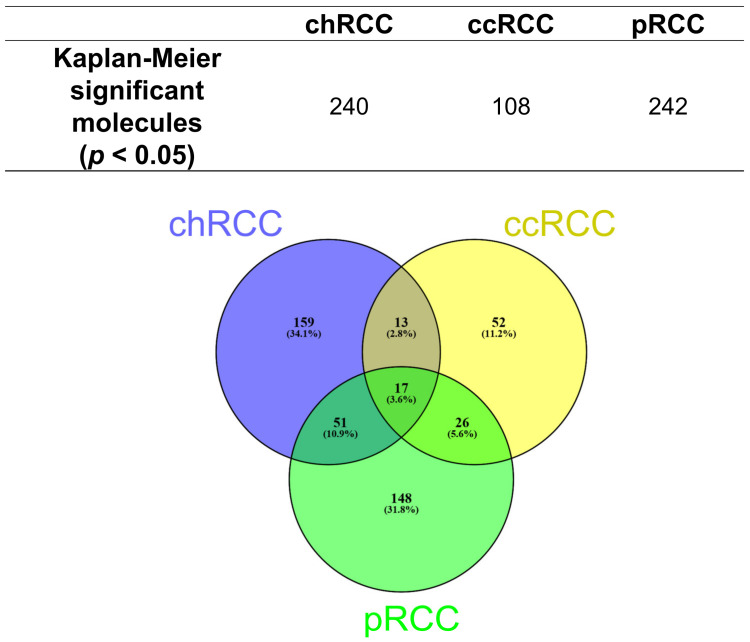
Kaplan–Meier significant metabolites in chRCC, pRCC and ccRCC.

**Figure 2 cancers-14-01763-f002:**
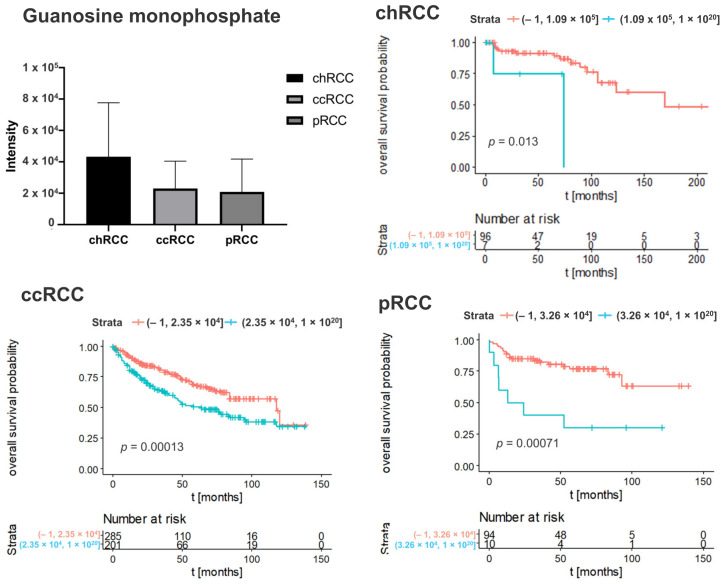
Kaplan–Meier curve of guanosine monophosphate in different subtypes of renal tumor. Blue lines indicate survival in patients with high intensity of the respective mass. Red lines indicate survival in patients with low intensity of the respective mass. Statistical significance testing was performed using the Kruskal–Wallis test (alpha = 0.05).

**Figure 3 cancers-14-01763-f003:**
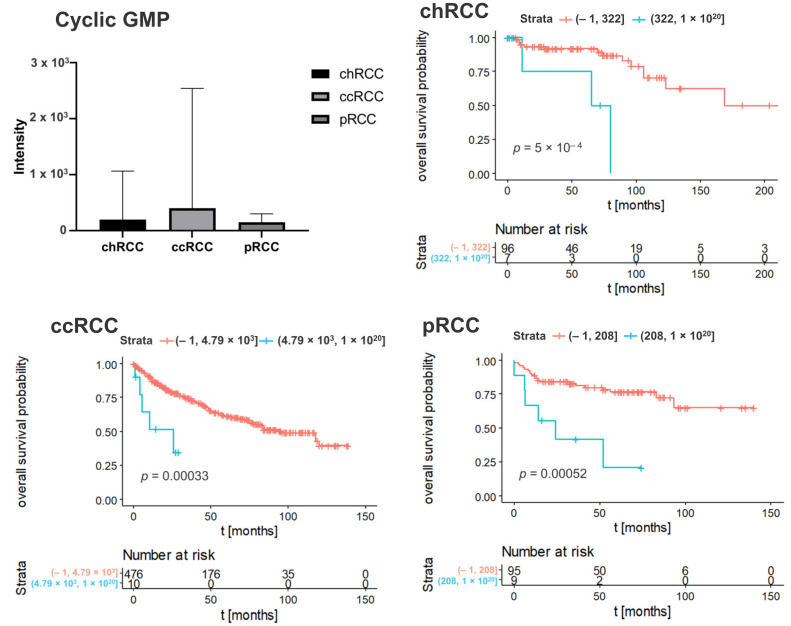
Kaplan–Meier curve of cyclic GMP in different subtypes of renal tumor. Blue lines indicate survival in patients with high intensity of the respective mass. Red lines indicate survival in patients with low intensity of the respective mass. Statistical significance testing was performed using the Kruskal–Wallis test (alpha = 0.05).

**Figure 4 cancers-14-01763-f004:**
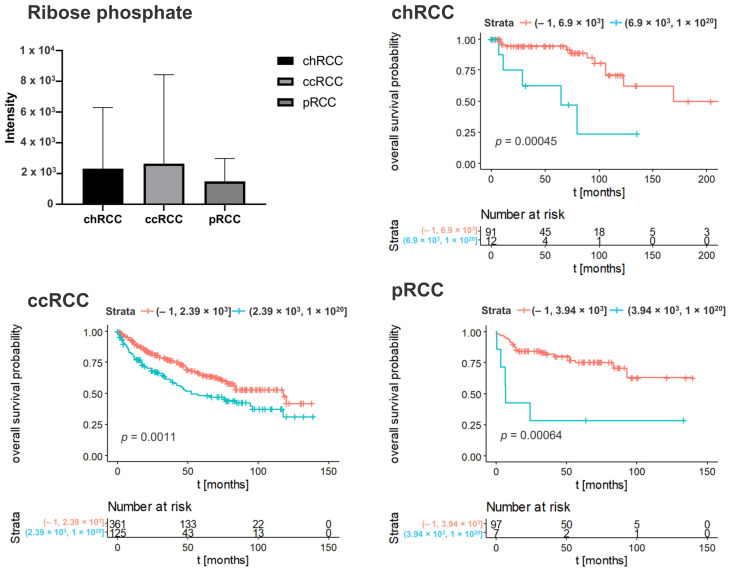
Kaplan–Meier curve of ribose phosphate in different subtypes of renal tumor. Blue lines indicate survival in patients with high intensity of the respective mass. Red lines indicate survival in patients with low intensity of the respective mass. Statistical significance testing was performed using the Kruskal–Wallis test (alpha = 0.05).

**Figure 5 cancers-14-01763-f005:**
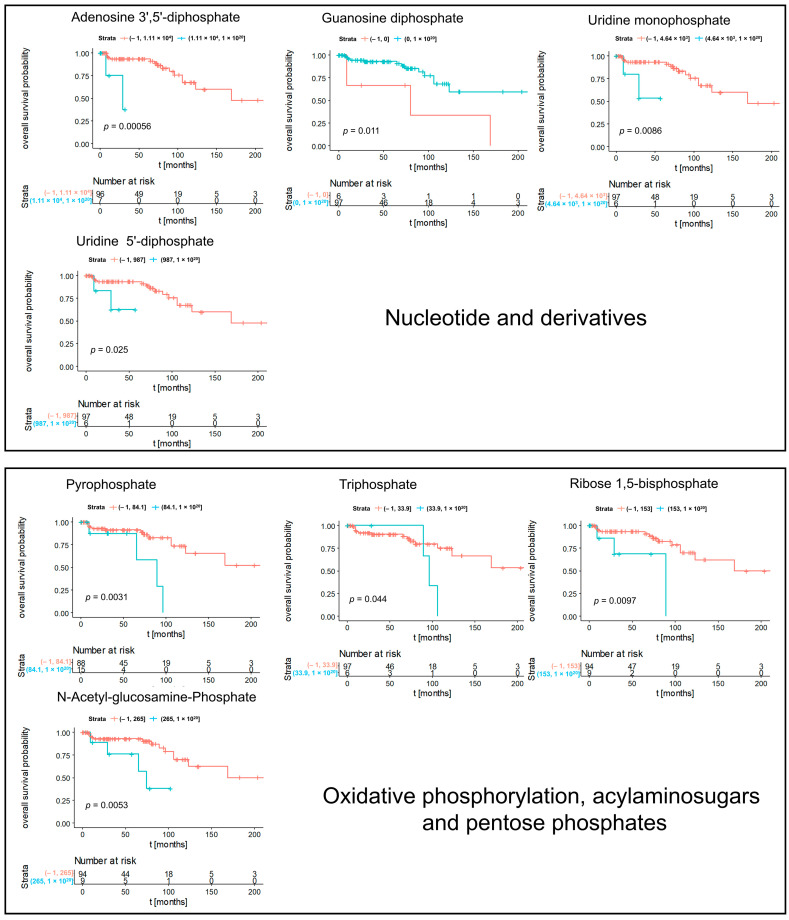
Kaplan–Meier significant metabolites specific for subtype chRCC. Blue lines indicate survival in patients with high intensity of the respective mass. Red lines indicate survival in patients with low intensity of the respective mass.

**Figure 6 cancers-14-01763-f006:**
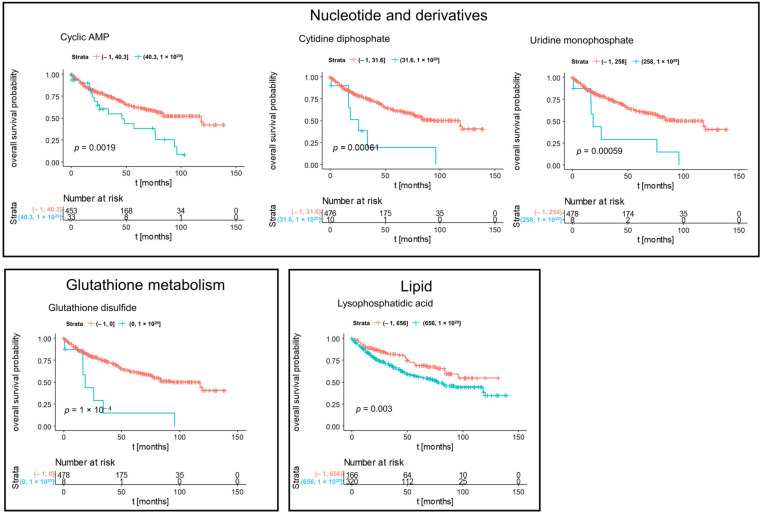
Kaplan–Meier significant metabolites specific for subtype ccRCC. Blue lines indicate survival in patients with high intensity of the respective mass. Red lines indicate survival in patients with low intensity of the respective mass.

**Figure 7 cancers-14-01763-f007:**
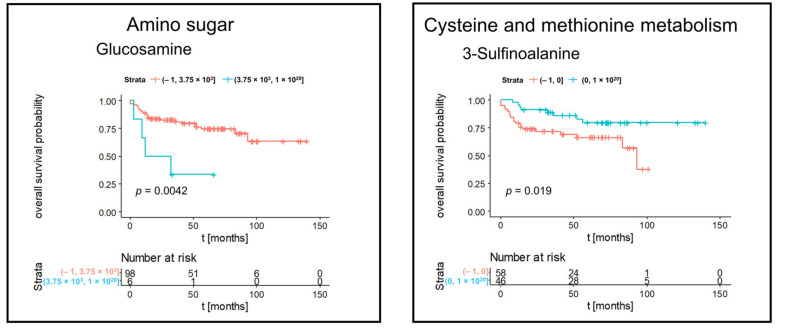
Kaplan–Meier significant metabolites specific for subtype pRCC. Blue lines indicate survival in patients with high intensity of the respective mass. Red lines indicate survival in patients with low intensity of the respective mass.

**Table 1 cancers-14-01763-t001:** Clinical and pathological characteristics.

Patient Characteristics	*n* = 782
Age median (range) (years)	27–88 (64)
Gender	
Male	269 (39.4%)
Female	413 (60.6%)
ISUP Grade	
Grade 1	25 (4.1%)
Grade 2	241 (39.4%)
Grade 3	191 (31.3%)
Grade 4	154 (25.2%)
Pathological stage	
pT1	417 (53.6%)
pT2	90 (11.6%)
pT3	231 (29.7%)
pT4	10 (1.3%)
pN+	26 (15.0%)
pM+	4 (36.4%)
Subtype	
chRCC	108 (13.8%)
ccRCC	552 (70.6%)
ppRCC	122 (15.6%)
Survival (Dead/Alive)	214 (31.6%)/464 (68.4%)
Overall survival median (months)	36

## Data Availability

The datasets used and/or analyzed during the current study are available from the corresponding author on reasonable request.

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
