# Peer review of "MALDI Mass Spectrometry Imaging—Prognostic Pathways and Metabolites for Renal Cell Carcinomas"

_cancers, 2022, doi:10.3390/cancers14071763_

Round 1
Reviewer 1 Report
The manuscript is very interesting and relevant to the field. The manuscript is well-written and there are only a few areas that are slightly confusing or could be written better. My biggest criticism is more numbers need to be given throughout the manuscript.
Table 1 – They should create three more tables that describe the clinical and pathological variables of each subtype (ccRCC, chRCC, and ppRCC).
For all the Kaplan Meier curves shown, the numbers at risk per time point per group should be given below the table. Also, the cutoff values for the high (red line) and low (blue line) intensity need to be larger in all Kaplan Meier curve figures. I can barely read them in many of the figures. I would also describe the cutoff values in the figure legends.
Language issues
Line 137
Ribose phosphate levels, an important intermediate … pathway, is related to a negative patient prognosis (Figure 4).
Line 179
The diagnostic power of MALDI .. has already been shown in RCC.
Line 184
I would spell out oncocytoma the first time you use it.
Line 203
Metabolic differences associated with kidney origin could occur for chRCC that originates from intercalated cells of the collecting duct but, ccRCC and ppRCC metabolic variability are probably due to other causes because both subtypes originate from the proximal tubules.
Author Response
Please see the atta
Reviewer 1:
The manuscript is very interesting and relevant to the field. The manuscript is well-written and there are only a few areas that are slightly confusing or could be written better. My biggest criticism is more numbers need to be given throughout the manuscript.
Table 1 – They should create three more tables that describe the clinical and pathological variables of each subtype (ccRCC, chRCC, and ppRCC).
Response:
We prepared Supplementary Tables 1-3 to describe of the clinical data of each subtype (ccRCC, chRCC, and ppRCC).
For all the Kaplan Meier curves shown, the numbers at risk per time point per group should be given below the table. Also, the cutoff values for the high (red line) and low (blue line) intensity need to be larger in all Kaplan Meier curve figures. I can barely read them in many of the figures. I would also describe the cutoff values in the figure legends.
Response:
We have added cut off values and patient survival table of the best and worst survivor cohorts of the Kaplan Meier curves in the Figurers. We increased font size in the Figures.
Language issues
Line 137
Ribose phosphate levels, an important intermediate … pathway, is related to a negative patient prognosis (Figure 4).
Response:
We modified the text as followings:
“Ribose phosphate is an important intermediate metabolite in the pentose phosphate pathway and in the purine metabolism pathway. Based on the Kaplan-Maier survival analysis, ribose phosphate was associated with unfavorable patient outcome.”
Line 179
The diagnostic power of MALDI .. has already been shown in RCC.
Response:
We modified the text as followings:
“The diagnostic power of MALDI for tumor discrimination of several entities have already been shown.”
Line 184
I would spell out oncocytoma the first time you use it.
Response:
We modified the text as followings:
“Zhang et al. demonstrated that MALDI enables discrimination of renal Oncocytoma (rO) from RCC subtypes and normal kidney tissue in 81 banked frozen human tissue samples.”
Line 203
Metabolic differences associated with kidney origin could occur for chRCC that originates from intercalated cells of the collecting duct but, ccRCC and ppRCC metabolic variability are probably due to other causes because both subtypes originate from the proximal tubules.
Response:
We modified the text accordingly:
“Metabolic differences associated with kidney origin could occur for chRCC that originates from intercalated cells of the collecting duct but, ccRCC and ppRCC metabolic variability are probably due to other causes because both subtypes originate from the proximal tubules.”
chment.
Reviewer 2 Report
In the manuscript "MALDI Mass Spectrometry Imaging - Prognostic Pathways and Metabolites for Renal Cell Carcinomas", the authors use MALDI IMS to analyze almost 800 samples for various metabolites. In general, I find the research excellent and think it will be a wonderful addition to the community. I also applaud the authors for their work.
I have some comments that I think would be helpful for the authors to address:
- The mass range the authors acquire on the FT-ICR is fairly broad (75-1000) and, generally, you take a hit in sensitivity and mass accuracy when the window is that high for a portion of the mass range. The calibration the authors do does not appear sufficient to correct this and the ppm error they accept (4 ppm) is too high for this system. In general, ppm error should be below 1 ppm, but 2 ppm would be reasonable.
- The authors use strange adducts for negative mode. Unfortunately, I did not see an SI or mass list of all the species the authors detected with their respective ppm errors. It's hard to determine the validity of these assignments. Also, was any validation (tandem ms, LC MS, etc.) performed for the assignments.
- Finally, the authors choose MALDI IMS to perform these experiments on but don't seem to incorporate the spatial component. Arguably, the authors could have performed a more in-depth analysis with a bulk approach and this would have given better information. Can the spatial component be incorporated or discussed to validate its use? I also am curious about laser settings and spatial resolution.
Author Response
Reviewer 2:
In the manuscript "MALDI Mass Spectrometry Imaging - Prognostic Pathways and Metabolites for Renal Cell Carcinomas", the authors use MALDI IMS to analyze almost 800 samples for various metabolites. In general, I find the research excellent and think it will be a wonderful addition to the community. I also applaud the authors for their work.
I have some comments that I think would be helpful for the authors to address:
- The mass range the authors acquire on the FT-ICR is fairly broad (75-1000) and, generally, you take a hit in sensitivity and mass accuracy when the window is that high for a portion of the mass range. The calibration the authors do does not appear sufficient to correct this and the ppm error they accept (4 ppm) is too high for this system. In general, ppm error should be below 1 ppm, but 2 ppm would be reasonable.
Response:
In this study, we used external calibration with L-arginine and internal lock mass calibration with 9-AA matrix ion signal (m/z 193.077122) minimizing scan-to-scan (pixel-to-pixel) variations during the MALDI-MSI measurement. The mass accuracy of each pixel is within sub ppm using the high mass resolution FT-ICR mass spectrometer.
For the metabolite database annotation, we chose 4 ppm tolerance which is consistent with previous publications and often used in MALDI imaging experiments (PMID: 32278304, PMID: 35184396, PMID: 20388918, PMID: 23685962). The database annotation is based on merged peaks of all spectrum after resampling, noise filtering and alignment (Detailed procedures please refer to PMID: 32278304 and PMID: 27414759). We decided to use 4 ppm mass tolerance to fit to the window of merge and resampling processes for peak picking and peak alignment. Supplementary Table 5 indicate most of the annotations are within 2 ppm accuracy.
- The authors use strange adducts for negative mode. Unfortunately, I did not see an SI or mass list of all the species the authors detected with their respective ppm errors. It's hard to determine the validity of these assignments. Also, was any validation (tandem ms, LC MS, etc.) performed for the assignments.
Response:
We thank the reviewer for this comment and the associated suggestions.
Adducts including [M-H], [M-H-H2O], [M+Na-2H], [M+Cl] and [M+K-2H] are commonly used adduct type in negative mode in the Human Metabolome Database (PMID: 34986597) and in previous studies (PMID: 25965788, PMID: 32278304, PMID: 35184396, PMID: 20388918). We consider these conventional adducts in our study to ensure the annotation coverage during the database search.
This study is a non-targeted metabolomics study for comprehensive and systematical identification of wide range of metabolites from different metabolic pathways. It is an unbiased metabolomics approach for discovery and hypotheses generation. The most common approach used to address molecular complexity is the coupling of high mass-resolving power with high-accuracy mass analyzers using mass spectrometry imaging of ion sources. This combination allows the mass resolution of many isobaric ion species and the direct assignment of elemental compositions, providing insights into the specific identities of the detected molecules. Our study was performed exclusively using high mass resolution mass spectrometry. Based on the high mass resolution mass spectrometry outcomes, we were able to accurately match peaks of interest against a molecular database according to m/z values. Mass spectrometry peaks detected in all tissues were then submitted to metabolite databases for annotation based on m/z values. For further MS/MS based metabolite validation of specific metabolites, due to the limited amount of TMA material, we cannot perform tissue extraction for LC-MS/MS experiments. Although alternative annotations exist for some compounds, most are structural variants of the same compound or priorities can be assessed according to mass accuracy, adduct, or plausibility. Accordingly, we provided supplementary data summarized mass list, annotations, adducts, and ppm.
- Finally, the authors choose MALDI IMS to perform these experiments on but don't seem to incorporate the spatial component. Arguably, the authors could have performed a more in-depth analysis with a bulk approach and this would have given better information. Can the spatial component be incorporated or discussed to validate its use? I also am curious about laser settings and spatial resolution.
Response:
In our current study, we used Spatial Correlation Image Analysis (SPACiAL) pipeline to generate a reference image based on MALDI imaging files. The digitized and co-registered staining images are scaled to match the exact MALDI resolution and then converted into numerical data without loss of spatial resolution. The spatial component were incorporated to extract mean peak intensities of tumor regions for each patient and performed prognosis calculation. We provided an additional section in Materials and Methods parts describing the workflow of the imaging pipeline Spatial Correlation Image Analysis (SPACiAL).
We added laser settings and spatial resolution in the Materials and Methods part:
“Metabolites were detected in negative-ion mode on a 7 T Solarix XR FT-ICR mass spectrometer (Bruker Daltonik) equipped with a dual ESI-MALDI source and a SmartBeam-II Nd: YAG (355 nm) laser. Laser attenuation value is 15%. The laser operated with focus of small (84.2 %). The laser operated at a frequency of 1000 Hz utilizing 200 laser shots per pixel with a pixel resolution of 60 µm.”
Round 2
Reviewer 2 Report
Awesome! Excited to see this in press.
Author Response
Thank you very much!